# Quantile Risk Control: A Flexible Framework for Bounding the Probability of High-Loss Predictions

**Jake C. Snell**[*]
Princeton University
js2523@princeton.edu

**Thomas P. Zollo**
Columbia University
tpz2105@columbia.edu

**Zhun Deng**
Columbia University
zd2322@columbia.edu

**Toniann Pitassi**
Columbia University
& University of Toronto
toni@cs.columbia.edu

**Richard Zemel**
Columbia University
& University of Toronto
zemel@cs.columbia.edu

## Abstract

Rigorous guarantees about the performance of predictive algorithms are necessary in order to ensure their responsible use. Previous work has largely focused on bounding the expected loss of a predictor, but this is not sufficient in many risk-sensitive applications where the distribution of errors is important. In this work, we propose a flexible framework to produce a family of bounds on quantiles of the loss distribution incurred by a predictor. Our method takes advantage of the order statistics of the observed loss values rather than relying on the sample mean alone. We show that a quantile is an informative way of quantifying predictive performance, and that our framework applies to a variety of quantile-based metrics, each targeting important subsets of the data distribution. We analyze the theoretical properties of our proposed method and demonstrate its ability to rigorously control loss quantiles on several real-world datasets.

## 1 Introduction

Learning-based predictive algorithms have a great opportunity for impact, particularly in domains such as healthcare, finance and government, where outcomes carry long-lasting individual and societal consequences. Predictive algorithms such as deep neural networks have the potential to automate a plethora of manually intensive tasks, saving vast amounts of time and money. Moreover, when deployed responsibly, there is great potential for a better decision process, by improving the consistency, transparency, and guarantees of the system. As just one example, a recent survey found that a majority of radiologists anticipated that AI-based solutions will lead to fewer medical errors, less time spent on each exam, and more time spent with patients (Waymel et al., 2019). In order to realize such benefits, it is crucial that predictive algorithms are rigorously yet flexibly validated prior to deployment. The validation should be *rigorous* in the sense that it produces bounds that can be trusted with high confidence. It should also be *flexible* in several ways. First, we aim to provide bounds on a variety of loss-related quantities (risk measures): the bound could apply to the mean loss, or the 90th percentile loss, or the average loss of the 20% worst cases. Furthermore, the guarantees should adapt to the difficulty of the instance: easy instances should have strong guarantees, and as the instances become harder, the guarantees weaken to reflect the underlying uncertainty. We also want to go beyond simply bounding the performance of a fixed predictor and instead choose the optimal predictor from a set of candidate hypotheses that minimizes some target risk measure.

Validating the trustworthiness and rigor of a given predictive algorithm is a very challenging task. One major obstacle is that the guarantees output by the validation procedure should hold with respect to any unknown data distribution and across a broad class of predictors including deep neural

---

[*]Work done while at University of Toronto and Vector Institute.

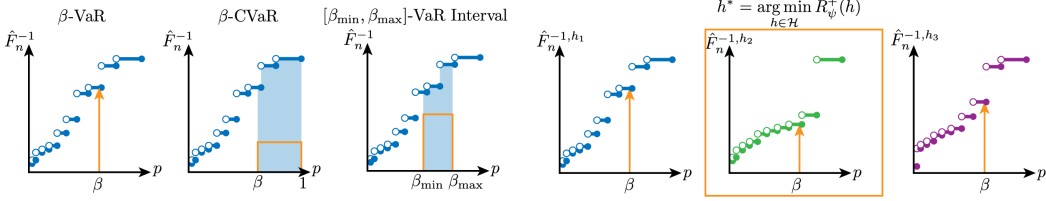

(a) Bounding multiple QBRMs with $\hat{F}_n^{-1}(p)$.     (b) Selecting the optimal $h^*$ with a target risk measure.

Figure 1: An overview of our quantile risk control framework. Given $n$ validation samples $X_1, \ldots, X_n$ drawn i.i.d. from the loss distribution with CDF $F$ we produce a upper confidence bound $\hat{F}_n^{-1}(p)$ on the true quantile function $F^{-1}(p) \triangleq \inf\{x : F(x) \geq p\}$. (a). The same bound on the quantile function can be used to bound multiple quantile-based risk measures. (b). An upper bound $\hat{F}_n^{-1,h}$ is computed for the quantile function of the loss distribution of each predictor $h \in \mathcal{H}$ and the one minimizing an upper confidence bound on the target risk measure $\hat{R}_\psi^+(h) = \int_0^1 \psi(p)\hat{F}_n^{-1,h}(p)\, dp$ is selected. Here the target measure is the $\beta$-VaR.

networks and complicated black-box algorithms. Recent work has built upon distribution-free uncertainty quantification to provide rigorous bounds for a single risk measure: the expected loss (Bates et al., 2021; Angelopoulos et al., 2021). However, to our knowledge there has been no work that unifies distribution-free control of a set of expressive risk measures into the same framework.

Our key conceptual advancement is to work with lower confidence bounds on the cumulative distribution function (CDF) of a predictor's loss distribution as a fundamental underlying representation. We demonstrate that a lower bound on the CDF can be converted to an upper bound on the quantile function. This allows our framework to seamlessly provide bounds for any risk measure that can be expressed as weighted integrals of the quantile function, known as quantile-based risk measures (QBRM) (Dowd & Blake, 2006). QBRMs are a broad class of risk measures that include expected loss, value-at-risk (VaR), conditional value-at-risk (CVaR) (Rockafellar & Uryasev, 2000), and spectral risk measures (Acerbi, 2002). Our approach inverts a one-sided goodness-of-fit statistic to construct a nonparametric lower confidence bound on the loss CDF for each candidate predictor. Furthermore, our confidence lower bounds hold simultaneously across an entire set of candidate predictors, and thus can be used as the basis for optimization of a target risk measure. For example, our approach can be used to choose a threshold or set of thresholds on the scores produced by a complicated black-box prediction algorithm. Figure 1 illustrates an overview of our framework.

We conduct experiments on real-world datasets where the goal is to tune the threshold of a deep neural network with respect to four representative risk measures. (1). The expected loss is the mean loss over the test distribution. (2). The $\beta$-VaR measures the maximum loss incurred on a specific quantile, after excluding a $1 - \beta$ proportion of high-loss outliers. (3). The $\beta$-CVaR measures the mean loss for the worst $1 - \beta$ proportion of the population. (4), Finally, the VaR-interval can be interpreted as optimizing an uncertain loss quantile that belongs to a known range, i.e., for a range of $\beta$ values. We compare various methods for controlling these risk measures, including a novel one that is tailored to the CVaR and VaR-interval settings, and show that our approach bounds the loss CDF well across all scenarios. We also demonstrate how our framework can be used to achieve fairness by equalizing a target risk measure across groups within the population.

Our work unifies the rigorous control of quantile-based risk measures into a simple yet expressive framework grounded in lower confidence bounds on the loss CDF. Unlike previous approaches which only target a single risk measure, our approach provides a flexible and detailed understanding of how a predictive algorithm will perform after deployment. We provide a family of bounds that hold for any quantile-based risk measure, even those not optimized for prior to deployment. Practitioners using our method can thus easily and confidently respond to inquiries from regulators and other key stakeholders, thereby building trust between organizations deploying predictive algorithms and the individuals whose lives are affected by them. This rigorous and flexible validation is an important step towards the responsible use of predictive algorithms in order to benefit society.

## 2 BACKGROUND & RELATED WORK

**Distribution-free uncertainty quantification.** This line of work originated with conformal prediction (Vovk et al., 2005; Shafer & Vovk, 2008), which seeks to produce *coverage* guarantees: *prediction sets* or *prediction intervals* that are guaranteed to contain the target output with high probability. Conformal prediction offers a lightweight algorithm that can be applied on top of complex predictors to obtain provable coverage guarantees. Recent work has generalized conformal prediction to bounding the expected loss with respect to more complex loss functions (Bates et al., 2021; Angelopoulos et al., 2021). Like these works, we also are interested in provable distribution-free guarantees for general loss functions. While previous work focuses on bounding the expected loss, part of our contribution lies in bounding the loss for quantiles of the data distribution. Angelopoulos & Bates (2022) offer an accessible overview of distribution-free uncertainty quantification.

**Empirical processes.** Several methods consider the relationship between the empirical and true CDF. The DKW inequality (Dvoretzky et al., 1956) generates CDF-based confidence bounds. Later variants (Massart, 1990; Naaman, 2021) provide tighter bounds or generalize to wider settings. The confidence bound provided by the DKW inequality is closely related to the Kolmogorov–Smirnov (KS) test (Massey, 1951) – a classic test of goodness of fit. The KS test has low power for detecting deviations in the tails and so other variance-weighted tests of goodness of fit address this, e.g., Anderson & Darling (1952); Cramér (1928). In order to handle extreme tails, Berk & Jones (1979) propose a statistic which looks for the most statistically significant deviation. Moscovich et al. (2016) and Moscovich (2020) study properties of the Berk-Jones statistics and provide methods for fast computation. To our knowledge, none of these techniques have been used as in our work, to produce bounds on quantiles of the loss distribution incurred by a predictor. For more background on quantile functions, the reader may refer to (Shorack, 2000, Chapter 7). For an overview of empirical processes, Shorack & Wellner (2009) is a comprehensive reference.

**Risk metrics.** A number of approaches have been used to estimate confidence intervals for estimated risk measures. Our focus is on a broad class of quantile-based risk measures (QBRMs), which includes VaR, CVaR, and spectral risk measures. The value-at-risk at the $\beta$ level ($\beta$-VaR) has long been used to express the risk associated with an investment and represents the maximum loss that the investment will incur with probability $\beta$. Conditional value-at-risk (CVaR) (Rockafellar & Uryasev, 2000) represents the mean loss incurred in the tail of the distribution. The work of Dowd (2010) inspires us; it presents a general approach for estimating confidence intervals for any type of quantile-based risk measure (QBRM). Here we construct confidence bounds on the loss CDF in order to bound such measures for the general problem of predictor selection.

## 3 PRELIMINARIES

In this section, we introduce our problem formulation and briefly discuss previous approaches that rigorously bound the expected loss. We then review quantile-based risk measures (QBRMs), an expressive class of risk measures that includes VaR, CVaR, and spectral risk measures.

### 3.1 PROBLEM SETUP & NOTATION

We assume the presence of a black-box predictor $h : \mathcal{Z} \to \hat{\mathcal{Y}}$ that maps from an input space $\mathcal{Z}$ to a space of predictions $\hat{\mathcal{Y}}$. We also assume a loss function $\ell : \hat{\mathcal{Y}} \times \mathcal{Y} \to \mathbb{R}$ that quantifies the quality of a prediction $\hat{Y}$ with respect to the target output $Y$. Let $(Z, Y)$ be drawn from an unknown data distribution $\mathcal{D}$ over $\mathcal{Z} \times \mathcal{Y}$ and define the random variable $X \triangleq \ell(h(Z), Y)$ to be the loss induced by $h$ on $\mathcal{D}$. Recall that the cumulative distribution function (CDF) of a random variable $X$ is defined as $F(x) \triangleq P(X \leq x)$. Our goal is to produce rigorous upper bounds on the risk $R(F)$ for a rich class of risk measures $R \in \mathcal{R}$, given a set of validation loss samples $X_{1:n} = \{X_1, \ldots, X_n\}$. For a detailed summary of the notations we used in this paper, please refer to Table 4 in Appendix A.

### 3.2 DISTRIBUTION-FREE CONTROL OF EXPECTED LOSS

Previous work has investigated distribution-free rigorous control of the expected loss. Recall that $F$ is the CDF of the loss r.v. $X$. Expected loss may be defined as $R(F) \triangleq \mathbb{E}[X]$. These methods

typically involve constructing a confidence region based on the sample mean $\bar{X} \triangleq \frac{1}{n}\sum_{i=1}^{n} X_i$ within which the true expected loss will lie with high probability. For example, suppose that $X \in [0,1]$ and let $\mu \triangleq \mathbb{E}[X]$. Hoeffding's inequality states that $P(\mu - \bar{X} \geq \varepsilon) \leq \exp(-2n\varepsilon^2)$. This may be inverted to provide an upper confidence bound on the expected loss: $P(R(F) \leq \hat{R}^+(X_{1:n})) \geq 1 - \delta$, where $\hat{R}^+(X_{1:n}) = \bar{X} + \sqrt{\frac{1}{2n}\log\frac{1}{\delta}}$. More sophisticated concentration inequalities, such as the Hoeffding-Bentkus inequality (Bates et al., 2021), have also been used to control the expected loss. Methods in this class differ primarily in how they achieve simultaneous control of multiple predictors: RCPS (Bates et al., 2021) does so by assuming monotonicity of the loss with respect to a one-dimensional space of predictors and Learn then Test (LTT) (Angelopoulos et al., 2021) applies a Bonferroni correction to $\delta$ in order to achieve family-wise error rate control (Lehmann & Romano, 2008, Sec. 9.1).

### 3.3 QUANTILE-BASED RISK MEASURES

Unlike previous work that considers only expected loss, we consider a class of risk measures known as quantile-based risk measures (QBRMs) (Dowd & Blake, 2006). Recall that the quantile function is defined as $F^{-1}(p) \triangleq \inf\{x : F(x) \geq p\}$.

**Definition 1** (Quantile-based Risk Measure). *Let $\psi(p)$ be a weighting function such that $\psi(p) \geq 0$ and $\int_0^1 \psi(p)\,dp = 1$. The quantile-based risk measure defined by $\psi$ is*

$$R_\psi(F) \triangleq \int_0^1 \psi(p) F^{-1}(p)\,dp. \tag{1}$$

Several representative QBRMs are shown in Table 1. The expected loss targets the overall expectation across the data distribution, whereas the VaR targets the maximum loss incurred by the majority of the population excluding a $(1-\beta)$ proportion of the population as outliers. CVaR on the other hand targets the worst-off/highest loss tail of the distribution. Finally, the VaR-Interval is valuable when the precise cutoff for outliers is not known. Other QBRMs, e.g., spectral risk measures (Acerbi, 2002), can easily be handled by our framework, but we do not consider them here.

| Risk Measure | Weighting Function $\psi(p)$ |
|---|---|
| Expected loss | 1 |
| $\beta$-VaR | $\delta_\beta(p)$ |
| $\beta$-CVaR | $\psi(p) = \begin{cases} \frac{1}{1-\beta}, & p \geq \beta \\ 0, & \text{otherwise} \end{cases}$ |
| $[\beta_{\min}, \beta_{\max}]$-VaR-Interval | $\psi(p) = \begin{cases} \frac{1}{\beta_{\max}-\beta_{\min}}, & \beta_{\min} \leq p \leq \beta_{\max} \\ 0, & \text{otherwise} \end{cases}$ |

Table 1: Several quantile-based risk measures and their corresponding weight functions (see Definition 1). See Figure 1 for visualizations. The Dirac delta function centered at $\beta$ is denoted by $\delta_\beta$.

## 4 QUANTILE RISK CONTROL

In this section we introduce our framework *Quantile Risk Control* (QRC) for achieving rigorous control of quantile-based risk measures (QBRM). QRC inverts a one-sided goodness-of-fit test statistic to produce a lower confidence bound on the loss CDF. This can subsequently be used to form a family of upper confidence bounds that hold for any QBRM. More formally, specify a confidence level $\delta \in (0,1)$ and let $X_{(1)} \leq \ldots \leq X_{(n)}$ denote the order statistics of the validation loss samples. QRC consists of the following high-level steps:

1. Choose a one-sided test statistic of the form $S \triangleq \min_{1 \leq i \leq n} s_i(F(X_{(i)}))$, where $F$ is the (unknown) CDF of $X_1, \ldots, X_n$.

2. Compute the critical value $s_\delta$ such that $P(S \geq s_\delta) \geq 1 - \delta$.

3. Construct a CDF lower confidence bound $\hat{F}_n$ defined by coordinates $(X_{(1)}, b_1), \ldots, (X_{(n)}, b_n)$, where $b_i \triangleq s_i^{-1}(s_\delta)$.

4. For any desired QBRM defined by weighting function $\psi$, report $R_\psi(\hat{F}_n)$ as the upper confidence bound on $R_\psi(F)$.

We show below that in fact $P(R_\psi(F) \leq R_\psi(\hat{F}_n)) \geq 1 - \delta$ for any QBRM weighting function $\psi$. If the target weighting function is known a priori, this information can be taken into account when choosing the statistic $S$. QRC can also be used to bound the risk of multiple predictors simultaneously by setting the critical value to $\delta' \triangleq \delta/m$, where $m$ is the number of predictors.

**Novelty of QRC.** There are two primary novel aspects of QRC. The first is the overall framework that uses nonparametric CDF lower confidence bounds as an underlying representation to simultaneously bound any QBRM. Previous works in distribution-free uncertainty quantification have targeted the mean and VaR, but to our knowledge none have focused on either the CVaR or VaR-interval. On the other hand, Dowd (2010) shows how to bound QBRMs given a CDF estimate but does not provide the tools to bound the CDF from a finite data sample. QRC brings these two ideas together by leveraging modern goodness-of-fit test statistics to produce high-quality nonparametric CDF lower bounds from finite data. The second novel aspect of QRC is our proposed forms of a truncated Berk-Jones statistic (Appendix C), which refocus the statistical power of standard Berk-Jones to target either the CVaR (one-sided truncation) or VaR-interval (two-sided truncation).

## 4.1 CDF LOWER BOUNDS ARE RISK UPPER BOUNDS

We begin by establishing that a lower bound $G$ on the true loss CDF $F$ incurred by a predictor can be used to bound $R_\psi(F)$ for any quantile-based risk measure weighting function. However, when we only have access to a finite sample $X_{1:n}$, it need to consider lower confidence bounds (LCB) on the CDF. For CDFs $F$ and $G$, let $F \succeq G$ denote $F(x) \geq G(x)$ for all $x \in \mathbb{R}$. We call $\hat{F}_n$ a $(1 - \delta)$-CDF-LCB if for any $F$, $P(F \succeq \hat{F}_n) \geq 1 - \delta$, where $\hat{F}_n$ is a function of $X_{1:n} \sim^{iid} F$.

**Theorem 1.** *Let $F$ and $G$ be CDFs such that $F \succeq G$. Then $R_\psi(F) \leq R_\psi(G)$ for any weighting function $\psi(p)$ as defined in Definition 1. Moreover, suppose that $\hat{F}_n$ is a $(1 - \delta)$-CDF-LCB. Then $P(R_\psi(F) \leq R_\psi(\hat{F}_n)) \geq 1 - \delta$.*

## 4.2 INVERTING GOODNESS-OF-FIT STATISTICS TO CONSTRUCT CDF LOWER BOUNDS

Our technique centers around producing a set of lower confidence bounds on the uniform order statistics, and using these to bound $F$. Let $U_1, \ldots, U_n \sim^{iid} U(0, 1)$ and let $U_{(1)} \leq \ldots \leq U_{(n)}$ denote the corresponding order statistics. Consider a one-sided minimum goodness-of-fit (GoF) statistic of the following form:

$$S \triangleq \min_{1 \leq i \leq n} s_i(U_{(i)}), \tag{2}$$

where $s_1, \ldots, s_n : [0, 1] \to \mathbb{R}$ are right continuous monotone nondecreasing functions. This statistic can be used to determine a set of constraints on a CDF as follows.

**Theorem 2.** *Let $s_\delta = \inf_r\{r : P(S \geq r) \geq 1 - \delta\}$, where $\delta \in (0, 1)$ and $S$ is defined above. Let $X_1, \ldots, X_n \sim^{iid} F$, where $F$ is an arbitrary CDF and let $X_{(1)} \leq \ldots \leq X_{(n)}$ be the corresponding order statistics. Then $P(\forall i : F(X_{(i)}) \geq s_i^{-1}(s_\delta)) \geq 1 - \delta$, where $\forall s \in \mathbb{R}$, $s_i^{-1}(s) \triangleq \inf_u\{u : s_i(u) \geq s\}$ is the generalized inverse of $s_i$.*

We use the constraints given by Theorem 2 to form a $(1-\delta)$-CDF-LCB via conservative completion of the CDF. An illustration of the process can be found in Figure 2.

## 4.3 CONSERVATIVE CDF COMPLETION

Here we show how to use constraints generated by Theorem 2 to form a lower bound on $F$ via the so-called conservative completion of the CDF (Learned-Miller & Thomas, 2020). This defines the "lowest lower bound" of $F$ given constraints.

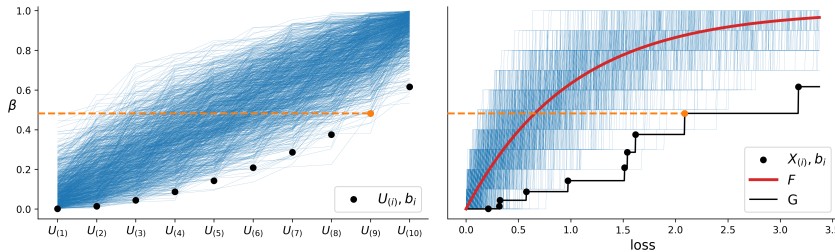

Figure 2: Example illustrating the construction of a distribution-free CDF lower bound by bounding order statistics. On the left, order statistics $U_{(1)}, \ldots, U_{(n)}$ are drawn from a uniform distribution. On the right, order statistics $X_{(1)}, \ldots, X_{(n)}$ are drawn from an arbitrary distribution (in this case a gamma distribution), and the corresponding Berk-Jones CDF lower confidence bound is shown as $G$. Our distribution-free method gives bound $b_i$ on each sorted order statistic such that the bound depends only on $i$, as illustrated in the plots for $i = 9$ (shown in orange). On the left, 1000 realizations of $U_{(1)}, \ldots, U_{(n)}$ are shown in blue. On the right, 1000 empirical CDFs $\bar{F}_n(x) \triangleq \frac{1}{n} \sum_{i=1}^{n} \mathbb{1}\{X_i \leq x\}$ are shown in blue, and the true CDF $F$ is shown in red.

**Theorem 3.** *Let $F$ be an arbitrary CDF satisfying $F(x_i) \geq b_i$ for all $i \in \{1, \ldots, n\}$, where $x_1 \leq \ldots \leq x_n$ and $0 \leq b_1 \leq \ldots \leq b_n < 1$. Let us denote $\mathbf{x} = (x_1, x_2, \cdots, x_n)$ and $\mathbf{b} = (b_1, b_2, \cdots, b_n)$ and let $x^+ \in \mathbb{R} \cup \infty$ be an upper bound on $X \sim F$, i.e. $F(x^+) = 1$. Let $G_{(\mathbf{x}, \mathbf{b})}(x)$ be defined as $0$ if $x < x_1$; $b_i$ if $x_i \leq x < x_{i+1}$ for $i \in \{1, 2, \cdots, n-1\}$; $b_n$ if $x_n \leq x < x^+$; and $1$ if $x^+ \leq x$. Then, $F \succeq G_{(\mathbf{x}, \mathbf{b})}$.*

**Corollary 1.** *Let $S$ be a test statistic defined as equation 2 and recall $s_\delta = \inf_r\{r : P(S \geq r) \geq 1 - \delta\}$. Let $X_1, \ldots, X_n \sim^{iid} F$, where $F$ is an arbitrary CDF and let $X_{(1)} \leq \ldots \leq X_{(n)}$ be the corresponding order statistics. Then with probability at least $1 - \delta$, $F \succeq G_{(X_{(1:n)}, s_{1:n}^{-1}(s_\delta))}$, where $X_{(1:n)} \triangleq (X_{(1)}, \ldots, X_{(n)})$ and $s_{1:n}^{-1}(s_\delta) \triangleq (s_1^{-1}(s_\delta), \ldots, s_n^{-1}(s_\delta))$.*

Thus we now have a way to construct a lower confidence bound on $F$ given the order statistics of a finite sample from $F$ that is defined by a set of functions $s_1, \ldots, s_n$. In other words, $\hat{F}_n \triangleq G_{(X_{(1:n)}, s_{1:n}^{-1}(s_\delta))}$ is a $(1 - \delta)$-CDF-LCB and can be used to select a predictor to minimize a risk measure as in Section 4.4. We also refer to $s_\delta$ as the *critical value* and it can be computed efficiently by bisection using software packages, e.g. Moscovich (2020), that compute the one-sided non-crossing probability.

Several standard tests of uniformity may be viewed as instances of the minimum-type statistic $S$ defined in equation 2. For example, the *one-sided Kolmogorov-Smirnov (KS)* statistic and the Berk-Jones statistic, which achieves greater tail sensitivity. In later sections, we use the term *Order Stats* to refer to a modified Berk-Jones statistic where only a single order statistic is used. This method along with some classical ones (e.g. DKW inequality) can be adapted to handle a VaR interval combined with Bonferroni correction. Please see more detailed discussion in Appendix B.

## 4.4 Bounding Multiple Predictors Simultaneously

We now suppose that we have a finite set of predictors $\mathcal{H}$ defined by a set of functions applied to the output of a complex algorithm such as a deep neural network. Let $\phi$ be an arbitrary black box function and let $t_1, \ldots, t_m$ be a set of functions, e.g. different thresholding operations. Then the set of predictors is $\mathcal{H} = \{t_1 \circ \phi, \ldots, t_m \circ \phi\}$. The loss r.v. for a random predictor $h \in \mathcal{H}$ is denoted by $X^h \sim F^h$ and the corresponding validation loss samples are $X_{1:n}^h$.

**Theorem 4.** *Suppose that $\hat{F}_n$ is a $(1 - \delta/|\mathcal{H}|)$-CDF-LCB. Then $P(\forall h \in \mathcal{H} : F^h \succeq \hat{F}_n^h) \geq 1 - \delta$.*

Theorem 4 implies that we can simultaneously produce a lower confidence bound on each predictor's loss function with probability at least $1 - \delta$, given that $\hat{F}_n$ is a $(1 - \delta/|\mathcal{H}|)$-CDF-LCB. From the set of predictors $\mathcal{H}$, our framework selects $h_\psi^*$ with respect to a target weighting function $\psi(p)$ as

$$h_\psi^* \triangleq \operatorname*{arg\,min}_{h \in \mathcal{H}} R_\psi(\hat{F}_n^h). \tag{3}$$

By Theorem 1, $P(R_{\psi'}(F^{h^*_\psi}) \leq R_{\psi'}(\hat{F}^{h^*_\psi}_n)) \geq 1 - \delta$, including $\psi' \neq \psi$. This means that we are able to bound any quantile-based risk metric, even if it was not targeted during optimization.

**Remark 1.** *One of our main contributions concerns novel goodness-of-fit statistics. We introduce two forms of a truncated Berk-Jones statistic that are useful for targeting specific ranges of quantiles. These are discussed in detail in Appendix C.*

## 5 EXPERIMENTS

We perform experiments to investigate our Quantile Risk Control framework as well as our particular novel bounds. First, we test the value of the flexibility offered by QRC by studying the consequences of optimizing for a given target metric when selecting a predictor. This motivates the need for strong bounds on a rich set of QBRMs, and thus we next examine the strength of our novel truncated Berk-Jones method for bounding quantities like the VaR Interval and CVaR. Finally, we present experimental results across the complete canonical set of metrics (Mean, VaR, Interval, and CVaR) in order to understand the effects of applying the full Quantitative Risk Control framework.

Our comparisons are made using the point-wise methods: Learn Then Test-Hoeffding Bentkus (**LttHB**); RCPS with a Waudby-Smith-Ramdas bound (**RcpsWSR**); **DKW**; and Order Statistics (**Order Stats**). We also compare goodness of fit-based methods: Kolmogorov-Smirnov (**KS**); Berk-Jones (**BJ**); and truncated Berk-Jones (**One-sided BJ, Two-sided BJ**); all as described in previous sections. Our experiments span a diverse selection of real-world datasets and tasks. First, we investigate our framework in the context of common image tasks using state of the art computer vision models and the MS-COCO (Lin et al., 2014) and CIFAR-100 (Krizhevsky, 2009) datasets. MS-COCO is a multi-label object detection dataset, where the presence of 80 object classes are detected and more than one class may be present in a single instance. CIFAR-100 is a single-label image recognition task, which we perform in the challenging zero-shot classification setting using CLIP (Radford et al., 2021). In addition, we conduct a natural language processing experiment using the Go Emotions (Demszky et al., 2020) dataset and a fine-tuned BERT model (Devlin et al., 2018), where the goal is to recognize emotion in text and a single instance may have multiple labels. Finally, we evaluate the interaction between our method and subgroup fairness using the UCI Nursery dataset (Dua & Graff, 2017), where applicants are ranked for admissions to school. While all methods are theoretically guaranteed, bounds on the true CDF or risk may be violated with some probability given by $\delta$. We analyze how these violations occur in Appendix E.6.4. Full details for all datasets, models, and experiments can be found in Appendix E. All experiments focus on producing prediction sets, as is typical in the distribution-free uncertainty quantification (DFUQ) setting (Vovk et al., 2005; Shafer & Vovk, 2008; Bates et al., 2021; Angelopoulos et al., 2021), and are run for 1000 random trials with $\delta = 0.05$. Code for our experiments is publicly available[1].

### 5.1 VALUE OF OPTIMIZING FOR TARGET METRIC

A main feature of the QRC framework is the flexibility to target a range of metrics, including: Expected-value (**Mean**); $\beta$-VaR (**VaR**); $[\beta_{\min}, \beta_{\max}]$-VaR-Interval (**VaR-Interval**); and **CVaR**. The results presented here demonstrate the importance of that flexibility by showing that in order to produce the best bound on a given risk measure, it is important to optimize for that metric when selecting a predictor.

Using the MS-COCO, CIFAR-100, and Go Emotions classification tasks, we record the guaranteed Mean, VaR, VaR Interval, and CVaR given by the Berk-Jones bound when targeting each of these risk measures to choose a predictor. The bound is applied on top of a pre-trained model that produces a per-class score, i.e. $W = \phi(Z)$, where $W \in \mathbb{R}^K$ and $K$ is the number of classes. A prediction set is formed by selecting all classes with a score at least some value $t$: $\hat{Y} = \{k : W_k \geq t\}$. Therefore a predictor is described by its corresponding threshold, and our algorithms choose the predictor that minimizes the target metric according to the bound produced by a given method. See Appendix E.7 for more details on the predictor space. Performance is measured using a balanced accuracy metric, which evenly combines sensitivity and specificity measures, where sensitivity encourages larger prediction set sizes and specificity encourages smaller set sizes (see Appendix E.2 for an exact description). Refer to Table 2 for MS-COCO results and Appendix Table 5 for CIFAR-100 and Go

---

[1]https://github.com/jakesnell/quantile-risk-control

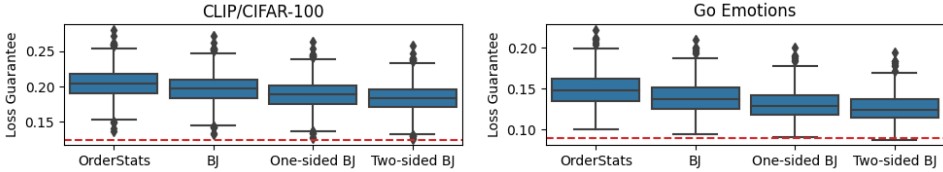

Figure 3: Results for bounding the quantities of the VaR for various methods. The average test loss over all trials is plotted in red.

Emotions results. Targeting a given metric in predictor selection gives the tightest bounds on that metric in all experiments. This highlights the significance of our QRC framework, which produces bounds over predictors for a diverse set of targeted risk measures such as the VaR Interval or CVaR.

| Dataset | Target | Mean | VaR | Interval | CVaR |
|---|---|---|---|---|---|
| MS-COCO | Mean | **0.088 ± 0.003** | 0.203 ± 0.014 | 0.211 ± 0.011 | 0.472 ± 0.017 |
| | VaR | 0.101 ± 0.006 | **0.179 ± 0.010** | 0.191 ± 0.011 | 0.460 ± 0.017 |
| | Interval | 0.100 ± 0.007 | 0.181 ± 0.011 | **0.189 ± 0.011** | 0.458 ± 0.017 |
| | CVaR | 0.107 ± 0.014 | 0.190 ± 0.016 | 0.197 ± 0.015 | **0.453 ± 0.015** |

Table 2: Results on MS-COCO illustrating trade-offs between targeting different metrics in predictor selection and the guarantees that can be issued for each metric given by the selected bound. We use the Berk-Jones bound, and bold results are best for a given metric.

## 5.2 TRUNCATED BERK-JONES VS. EXISTING METHODS

Next, we compare the strength of our truncated Berk-Jones bounds for the VaR-Interval and CVaR to existing methods for bounding these quantities for hypothesis selection. As opposed to the previous experiment, here we fix a single threshold prior to running the experiment, in order to isolate the strength of the bounds from the randomness induced by multiple predictors. Results are shown in Figure 3, Appendix Figure 5, and Figure 4, with further explanation below. The truncated Berk-Jones method consistently gives the best bounds, especially on the CVaR.

**Image and Text Classification.** Our novel truncated Berk-Jones bounds can be used to control the loss for a diverse range of tasks, datasets, and loss functions. Here we examine the strength of these bounds for bounding quantities of the VaR on multi-label image (MS-COCO) and text (Go Emotions) classification tasks, as well as single-label zero-shot image classification (CIFAR-100). For MS-COCO object detection, VaR-Interval is evaluated in $[0.85, 0.95]$, and CVaR is calculated with $\beta = 0.9$. For zero-shot classification on CIFAR-100 and emotion recognition in Go Emotions, the VaR-Interval is bounded in $[0.6, 0.9]$. Results are shown in Figure 3 and Appendix Figure 5. Across all tasks and datasets, the modified Berk-Jones bound consistently gives a loss guarantee that is lowest, and thus closest to the actual loss for this fixed predictor.

**Fair Classification.** Williamson & Menon (2019) proposed CVaR as a notion of fairness that can be viewed as the worst-case mean loss for subgroups of a particular size. Thus we consider a fairness problem, and show how our framework can be applied to generate a predictor that optimizes CVaR, or worst-case outcomes, individually with respect to each group according to some protected attribute. We employ the UCI Nursery dataset (Dua & Graff, 2017), a multi-class classification problem where categorical features about families are used to assign a predicted rank to an applicant $Y \in \{1, ..., 5\}$ ranging from "not recommended" ($Y = 1$) to "special priority" ($Y = 5$). We measure a class-dependent weighted accuracy loss. Our sensitive attribute is a family's financial status. Applicants are separated based on this sensitive attribute, and we apply the methods to and evaluate each group ("convenient" and "inconvenient") separately. Exact specifications of the loss function, dataset, and other experiments details can be found in Appendix E.5. Results for the fair classification experiment are reported in Figure 4, where Group 1 have convenient financial standing and Group 2 have inconvenient. Applying the truncated Berk-Jones bound individually to each group leads to the best CVaR among all methods for both groups.

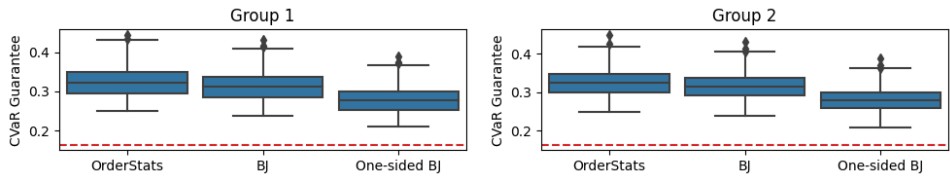

Figure 4: Results for bounding the CVaR for each group in Nursery with $\beta = 0.9$.

## 5.3 MS-COCO EXPERIMENTS ON ALL RISK MEASURES

For our final experiment we compare the performance of various bounding methods across the full canonical set of metrics (Mean, VaR, Interval, and CVaR) when selecting a predictor. We perform this experiment using MS-COCO and a balanced accuracy risk measure, and apply a Bonferroni correction to $\delta$ according to the size of the predictor space.

Results bounding the CVaR and VaR Interval are reported in Table 3, while all results are shown in Appendix Table 6. Tables include the target metric used to choose a predictor, bounding method, guarantee returned on the validation set, and loss quantity observed on the test set. As expected, RcpsWSR gives the tightest bound on the Mean, while the point-wise Order Statistics method outperforms the GoF-based methods on bounding a single VaR metric. When bounding a VaR-Interval or the CVaR, which is a focus of our work, the truncated Berk-Jones bounds achieves the lowest loss guarantee. In general, the best guarantees have best actual loss, although the difference can be small (or none) as all methods may choose the same or similar predictors around that which minimizes the target risk measure on the validation set.

| | Method | Guarantee | Actual | | Method | Guarantee | Actual |
|---|---|---|---|---|---|---|---|
| CVaR | DKW | 1.0 ± 0.0 | 0.500 ± 0.000 | Interval | DKW | 0.744 ± 0.005 | 0.146 ± 0.012 |
| | OrderStats | 0.454 ± 0.015 | 0.208 ± 0.011 | | OrderStats | 0.189 ± 0.011 | 0.138 ± 0.007 |
| | KS | 0.971 ± 0.002 | 0.228 ± 0.022 | | KS | 0.581 ± 0.008 | 0.144 ± 0.013 |
| | BJ | 0.453 ± 0.015 | 0.208 ± 0.011 | | BJ | 0.189 ± 0.011 | 0.138 ± 0.007 |
| | One-sided BJ | **0.419 ± 0.015** | **0.207 ± 0.010** | | One-sided BJ | 0.183 ± 0.010 | **0.137 ± 0.006** |
| | | | | | Two-sided BJ | **0.182 ± 0.010** | **0.137 ± 0.006** |

Table 3: Results for experiments with MS-COCO Object Detection bounding CVaR and the VaR Interval. Best results are in bold.

## 6 DISCUSSION

Our experimental results (see Section 5 and Appendix E.6) show that: (1) in order to achieve a tight bound on a target risk measure, a predictor should be selected based on that risk measure; and (2) the truncated Berk-Jones bounds are more effective than previous methods for bounding the VaR-Interval and CVaR. While methods that focus on the mean or a particular $\beta$-VaR value can give a tighter bound when that target metric is known beforehand, a key advantage of Quantile Risk Control comes from the flexibility to change the target metric post hoc, within some range, and still maintain a valid bound. Though there are some ways to convert between these classes (e.g., Mean to Quantile with Markov's Inequality; CVaR is directly an upper bound on VaR), these are weak as shown by our experiments. This makes QRC well-suited to scenarios in which a stakeholder may want to issue a guarantee or several guarantees from a single predictor, or when an external auditor may change system reporting requirements. A natural extension of this work is to consider returning multiple predictors with our algorithm. In this case, while the target metric may not be known beforehand, we could produce multiple predictors that optimally bound specific points in the range, instead of a single predictor that gives the best bound on average over the range.

ETHICS STATEMENT

Some care needs to be taken to ensure that the guarantees under our framework are clear, in that they do not apply to every individual in the test distribution and only hold if test examples are drawn from the same distribution as validation examples.

REPRODUCIBILITY STATEMENT

Our code is publicly available on Github. For all experiments, datasets and splits are specified, and we use publicly available pre-trained models for feature extraction when possible.

ACKNOWLEDGMENTS

The authors would like to thank Marc-Etienne Brunet, Anastasios Angelopoulos, and anonymous reviewers for their useful comments. Resources used in preparing this research were provided, in part, by the Province of Ontario, the Government of Canada through CIFAR, and companies sponsoring the Vector Institute (https://vectorinstitute.ai/partners/). This project is also supported by NSERC and Simons Foundation Collaborative Research Grant 733782.

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

# A    NOTATION

| Symbol | Meaning |
| --- | --- |
| $\mathcal{Z}$ | Input space. |
| $\mathcal{Y}$ | Target output space. |
| $\hat{\mathcal{Y}}$ | Prediction space. |
| $h$ | A predictor mapping from $\mathcal{Z}$ to $\hat{\mathcal{Y}}$. |
| $\mathcal{D}$ | Data distribution over $\mathcal{Z} \times \mathcal{Y}$. |
| $(Z, Y)$ | Random variables representing input-target pairs drawn from $\mathcal{D}$. |
| $\ell(\hat{Y}, Y)$ | Loss incurred by predicting $\hat{Y} \in \hat{\mathcal{Y}}$ when the ground truth is $Y \in \mathcal{Y}$. |
| $X^h$ | Random variable representing $\ell(h(Z), Y)$. This is the loss incurred by applying predictor $h$ to input $Z$ when the ground truth is $Y$. The superscript $h$ may be dropped when the predictor being referred to is clear from context. |
| $F^h$ | The cumulative distribution function (CDF) of $X^h$, also referred to as the loss CDF, defined as $F^h(x) \triangleq P(X^h \leq x)$. |
| $F^{-1,h}$ | Quantile function defined as $F^{-1,h}(p) \triangleq \inf\{x : F^h(x) \geq p\}$. |
| $R(F)$ | Risk measure quantifying the risk associated with loss CDF $F$. |
| $\psi$ | Weighting function defining a quantile-based risk measure (see Definition 1). |
| $X_{1:n}$ | A sequence of loss random variables $X_1, \ldots, X_n$. |
| $X_{(i)}$ | The $i$-th order statistic of $X_{1:n}$. Order statistics satisfy $X_{(1)} \leq \ldots \leq X_{(n)}$. |
| $\succeq$ | $F \succeq G$ denotes $F(x) \geq G(x)$ for all $x \in \mathbb{R}$. |
| $\hat{F}_n^h$ | A lower confidence bound on $F^h$ as a function of $X_1^h, \ldots, X_n^h$. |
| $\hat{F}_n^{-1,h}$ | The upper confidence bound on $F^{-1,h}$ corresponding to $\hat{F}_n^h$. |

Table 4: Summary of the notation used in this paper.

# B    INTERPRETING STANDARD TESTS OF UNIFORMITY AS MINIMUM-TYPE STATISTICS $S$

Several standard tests of uniformity may be viewed as instances of the minimum-type statistic $S$ defined in equation 2. For example, the *one-sided Kolmogorov-Smirnov (KS)* statistic with the uniform as the null hypothesis can be expressed as $D_n^+ \triangleq \max_{1 \leq i \leq n} \left( \frac{i}{n} - U_{(i)} \right) = -\min_{1 \leq i \leq n} \left( U_{(i)} - \frac{i}{n} \right)$, which can be viewed as $D_n^+ = -S$, where $s_i(u) = u - \frac{i}{n}$. The Berk-Jones statistic achieves greater tail sensitivity by taking advantage of the fact that the marginal distribution of $U_{(i)}$ is $\text{Beta}(i, n - i + 1)$, which has lower variance in the tails. The *one-sided Berk-Jones (BJ)* statistic is defined as $M_n^+ \triangleq \min_{1 \leq i \leq n} I_{U_{(i)}}(i, n - i + 1)$, where $I_x(a, b)$ is the CDF of $\text{Beta}(a, b)$ evaluated at $x$. This can be recognized as $S$ where $s_i(u) = I_u(i, n - i + 1)$, the inverse of which can be efficiently computed using standard software packages.

Standard distribution-free one-sided confidence bounds for quantiles involve selecting the minimal order statistic that bounds the desired quantile level with high probability. We use the term *Order Stats* to refer to a modified Berk-Jones statistic where only a single order statistic is used. An analogous approach can be taken using the *DKW* inequality (see Appendix E.1 for details). These methods can be adapted to handle a VaR interval by simultaneously bounding multiple quantile levels, e.g. using a grid over the interval $[\beta_{\min}, \beta_{\max}]$, via Bonferroni correction.

# C    NOVEL TRUNCATED BERK-JONES STATISTICS

One of our main contributions concerns novel goodness-of-fit statistics. Here we introduce two forms of a truncated Berk-Jones statistic, targeting a range of quantiles. The first form targets a one-sided range of quantiles $[\beta_{\min}, 1)$. The key idea is to drop lower order statistics that do not affect the bound on $F^{-1}(\beta_{\min})$. We define the *truncated one-sided Berk-Jones* as $M_{n,k}^+ \triangleq \min_{k \leq i \leq n} I_{U_{(i)}}(i, n - i + 1)$, which can be realized by using $s_i(u) = I_u(i, n - i + 1)$ for $k \leq i \leq n$

and $s_i(u) = 1$ otherwise. For a given $\beta_{\min}$, define $k^*(\beta_{\min}) \triangleq \min\{k : s_k^{-1}(s_\delta^k) \geq \beta_{\min}\}$, where $s_\delta^k$ is the critical value of $M_{n,k}^+$. Bisection search can be used to compute $k^*$ and thus the inversion of $M_{n,k^*(\beta_{\min})}^+$ provides a CDF lower bound targeted at quantiles $\beta_{\min}$ and above. The *truncated two-sided Berk-Jones* variant targets a quantile interval $[\beta_{\min}, \beta_{\max}]$ and is defined as $M_{n,k,\ell}^+ \triangleq \min_{k \leq i \leq \ell} I_{U_{(i)}}(i, n - i + 1)$. It first computes $k^*(\beta_{\min})$ as in the one-sided case and then removes higher order statistics using the upper endpoint of $\ell^* \triangleq \min\{\ell : s_\ell^{-1}(s_\delta^{k^*,\ell}) \geq \beta_{\max}\}$, where $s_\delta^{k^*,\ell}$ is the corresponding critical value.

## D  PROOFS

### D.1  PROOF OF THEOREM 1

*Proof.* Consider $G^{-1}(p)$. By the definition of the quantile function, $G(G^{-1}(p)) \geq p$. By the relationship between $F$ and $G$, we therefore have $F(G^{-1}(p)) \geq G(G^{-1}(p)) \geq p$, where the first inequality follows from $F \succeq G$. Applying $F^{-1}$ to both sides yields $F^{-1}(F(G^{-1}(p))) \geq F^{-1}(p)$. But $x \geq F^{-1} \circ F(x)$ (see e.g. Proposition 3 on p. 6 of Shorack & Wellner (2009)) and thus $G^{-1}(p) \geq F^{-1}(p)$. Then we have $R_\psi(F) \leq R_\psi(G)$ by Definition 1. The event that $F \succeq \hat{G}(X_{1:n})$ implies that $R_\psi(F) \leq R_\psi(G)$ by Theorem 1. Since the antecedent holds with probability at least $1 - \delta$, so must the consequent. $\square$

### D.2  PROOF OF THEOREM 2

*Proof.* For readability, we first discuss the case when the CDF $F$ is continuous and strictly increasing. After that, we demonstrate how to extend our argument to a more general case.

**Continuous and strictly increasing $F$.**  If $F$ is continuous and strictly increasing, we know $F$ is bijective and the inverse function $F^{-1}(p)$ is well-defined. Moreover, for $X \sim F$, we have $F(X)$ is of the same distribution as uniform random variable $U$. That is because

$$P(F(X) \leq p) = P(X \leq F^{-1}(p)) = p.$$

In addition, by the monotonicity of $F$, $F$ preserves ordering, which means $F(X_{(i)})$ is of the same distribution of $U_{(i)}$ for all $i \in [n]$. So, we have

$$P\left(\forall i,\ F(X_{(i)}) \geq s_i^{-1}(s_\delta)\right) = P\left(\forall i,\ U_{(i)} \geq s_i^{-1}(s_\delta)\right).$$

Now recall that

$$S \triangleq \min_{1 \leq i \leq n} s_i(U_{(i)}).$$

Given the definition of $s_\delta$, we know that

$$P\left(S \geq s_\delta\right) \geq 1 - \delta.$$

In addition, by the definition $s_i^{-1}$, if $S \geq s_\delta$, then we have that $U_{(i)} \geq s_i^{-1}(s_\delta)$ for all $i \in \{1, \ldots, n\}$. Combined together, we have

$$P\left(\forall i,\ F(X_{(i)}) \geq s_i^{-1}(s_\delta)\right) = P\left(\forall i,\ U_{(i)} \geq s_i^{-1}(s_\delta)\right) \geq P\left(S \geq s_\delta\right) \geq 1 - \delta.$$

**General $F$.**  For general $F$, recall we define a generalized inverse function $F^{-1}(p) \triangleq \inf\{x : F(x) \geq p\}$. Notice that for a generalized inverse function $F^{-1}$, we have $F^{-1}(U)$ is of the same distribution as $X \sim F$. That is because

$$P(F^{-1}(U) \leq c) = P(U \leq F(c)) = F(c).$$

We want to remark that the above formula holds for general $F$ and the first equality directly follows from the definition of the generalized inverse function $F^{-1}$. Moreover, $F^{-1}$ preserves ordering (by definition of the quantile function), i.e. $F^{-1}(U_{(1)}) \leq \ldots \leq F^{-1}(U_{(n)})$. Thus, we have $X_{(i)}$ is of the same distribution as $F^{-1}(U_{(i)})$.

By Proposition 1, eq. 24 on p.5 of Shorack & Wellner (2009), we have that

$$F \circ F^{-1}(t) \geq t$$

for any $t \in [0,1]$ and any CDF $F$ (this is can obtained directly from the definition of $F^{-1}$ also can go beyond CDF, but we here only restrict ourselves in the case that $F$ is a CDF). Therefore we have

$$F(F^{-1}(U_{(i)})) \geq U_{(i)} \geq s_i^{-1}(s_\delta).$$

Combined with the fact that $X_{(i)}$ is of the same distribution as $F^{-1}(U_{(i)})$ for all $i \in [n]$, we have

$$P\left(\forall i, \ F(X_{(i)}) \geq s_i^{-1}(s_\delta)\right) = P\left(\forall i, \ F(F^{-1}(U_{(i)}) \geq s_i^{-1}(s_\delta)\right)$$
$$= P\left(\forall i, \ U_{(i)} \geq s_i^{-1}(s_\delta)\right) \geq P\left(S \geq s_\delta\right)$$
$$\geq 1 - \delta.$$

$\square$

### D.3 PROOF OF THEOREM 3

*Proof.* When $x < x_1$, $F(x) \geq 0$. For any $i \in \{1, 2, \cdots, n\}$, since $F(x_i) \geq b_i$, thus, for $x \in [x_i, x_{i+1})$, $F(x) \geq F(x_i) \geq b_i$ ($x_{n+1}$ is considered as $x^+$ here). Lastly, if $x \geq x^+$, $F(x) = 1$. Combined together, we have $F(x) \geq G_{(\mathbf{x},\mathbf{b})}(x)$. $\square$

### D.4 PROOF OF COROLLARY 1

*Proof.* Notice that by Theorem 3, if $\forall i, F(X_{(i)}) \geq s_i^{-1}(s_\delta)$, then we have $F(x) \geq G_{(X_{(1:n)}s_{1:n}^{-1}(s_\delta))}$, which implies

$$P\left(F(x) \geq G_{(X_{(1:n)},s_{1:n}^{-1}(s_\delta))}(x)\right) \geq P\left(\forall i, F(X_{(i)}) \geq s_i^{-1}(s_\delta)\right).$$

Then, by Theorem 2, $P\left(\forall i, F(X_{(i)}) \geq s_i^{-1}(s_\delta)\right) \geq 1 - \delta$. $\square$

### D.5 PROOF OF THEOREM 4

*Proof.* The claim follows by applying a union bound argument over $h \in \mathcal{H}$:

$$P(\exists h \in \mathcal{H} : F^h \not\preceq \hat{G}(X_{1:n}^h)) \leq \sum_{h \in \mathcal{H}} P(F^h \not\preceq \hat{G}(X_{1:n}^h))) \leq \delta.$$

$\square$

## E  EXPERIMENTAL DETAILS

### E.1  DKW METHOD FOR QUANTILE DISTRIBUTION-FREE UPPER CONFIDENCE BOUNDS

A distribution upper confidence bound on $F^{-1}(\beta)$ can be constructed by using a one-sided version of the DKW inequality (Dvoretzky et al., 1956; Massart, 1990). Define the empirical distribution function as $F_n(x) = \frac{1}{n} \sum_{i=1}^n \mathbb{I}\{X_i \leq x\}$. Then for every $\varepsilon \geq \sqrt{\frac{1}{2n} \log 2}$, by the DKW inequality,

$$P\left\{\sup_{x \in \mathbb{R}} F_n(x) - F(x) > \varepsilon\right\} \leq \exp(-2n\varepsilon^2).$$

If we let $\delta = \exp(-2n\varepsilon^2)$, then we have $\varepsilon = \sqrt{\frac{1}{2n} \log \frac{1}{\delta}}$ and the above condition holds for any $\delta \leq 1/2$.

**Proposition 1.** *Let $X_{(1)} \leq \ldots \leq X_{(n)}$ denote the order statistics of $X_1, \ldots, X_n$, where $X_i \sim F$ for all $i \in \{1, \ldots, n\}$. Let $k = \left\lceil n\left(\beta + \sqrt{\frac{1}{2n} \log \frac{1}{\delta}}\right)\right\rceil$. Then for $\beta + \sqrt{\frac{1}{2n} \log \frac{1}{\delta}} \leq 1$ and $\delta \leq 1/2$,*

$$P(F^{-1}(\beta) \leq X_{(k)}) \geq 1 - \delta.$$

*Proof.* Observe that $F_n(X_{(i)}) \geq \frac{i}{n}$ for $i \in \{1, \ldots, n\}$. Thus for $k$ defined above, we have

$$F_n(X_{(k)}) \geq \beta + \sqrt{\frac{1}{2n} \log \frac{1}{\delta}}.$$

But with probability at least $1 - \delta$, we have

$$F(x) \geq F_n(x) - \sqrt{\frac{1}{2n} \log \frac{1}{\delta}}$$

for all $x \in \mathbb{R}$. Thus $P(F(X_{(k)}) \geq \beta) \geq 1 - \delta \Rightarrow P(F^{-1}(\beta) \leq X_{(k)}) \geq 1 - \delta$, where the latter follows from the property that $F^{-1} \circ F(x) \leq x$ (see e.g. Proposition 3 on p. 6 of Shorack & Wellner (2009)). □

### E.2   DEFINITIONS OF LOSS FUNCTIONS

Balanced accuracy is computed as follows, where $\hat{Y}$ is the prediction set, $Y$ is the set of ground truth labels (which in this case will always be a single label), and $K$ is the number of classes:

$$L(\hat{Y}, Y) = 1 - \frac{1}{2}(\text{Sens}(\hat{Y}, Y) + \text{Spec}(\hat{Y}, Y)), \text{ where}$$

$$\text{Sens}(\hat{Y}, Y) = \frac{|\hat{Y} \cap Y|}{|Y|} \text{ and } \text{Spec}(\hat{Y}, Y) = \frac{K - |Y| - |\hat{Y} \setminus Y|}{K - |Y|}.$$

### E.3   MS-COCO EXPERIMENT DETAILS

MS-COCO (Lin et al., 2014) is a large object detection dataset with 80 different categories where an image may contain one or more classes of objects and thus may have one or more ground truth labels. Class scores are extracted using TResNet (Ridnik et al., 2021). Results are averaged over 1,000 trials, where in each trial 2,000 datapoints are randomly split into 500 validation and 1,500 test points. VaR is targeted with $\beta = 0.9$, while the Interval and CVaR target the ranges $[0.85, 0.95]$ and $[0.9, 1.0]$, respectively. KS and BJ bound the Mean by targeting the loss for the entire range $[0.0, 1.0]$. Point-wise methods (DKW, OrderStats) are used to bound intervals by calculating VaR upper bounds over a grid of $\beta$ values within the range of interest using a Bonferroni correction for the grid size. The grid size is 10 for the VaR Interval, and 50 for CVaR.

### E.4   CIFAR-100 AND GO EMOTIONS EXPERIMENT DETAILS

We use a multi-modal CLIP (Radford et al., 2021) embedding model in order to perform zero-shot image classification on the CIFAR-100 dataset using the method described in Radford et al. (2021). CIFAR-100 features images of 100 different classes, and we use the 10,000 examples available in the test split. Our feature extractor is the *ViT-B/32* version of CLIP available on Hugging Face (Wolf et al., 2020).

Go Emotions (Demszky et al., 2020) is a dataset of comments from Reddit where binary labels are given for 28 different fine-grained emotion categories and each comment may have one or more positive labels. 5,427 examples are drawn from the test set, and our base classifier is a *bert-base-uncased* model fine-tuned on the train split, again available on Hugging Face (Wolf et al., 2020).

Experiments are run for 1,000 trials with 500 validation examples randomly selected and the rest of the dataset used for testing. Balanced accuracy loss is bounded in the VaR-Interval $[0.6, 0.9]$ in both cases. For zero-shot CLIP, point-wise methods are run over a grid of 50 $\beta$ values, and for Go Emotions the grid size is also 50.

### E.5   FAIRNESS EXPERIMENT

We measure a class-varying weighted accuracy loss:

$$L(\hat{Y}, Y) = (1 - \mu_Y)(1 - \text{Spec}(\hat{Y}, Y)) + \mu_Y(1 - \text{Sens}(\hat{Y}, Y))$$

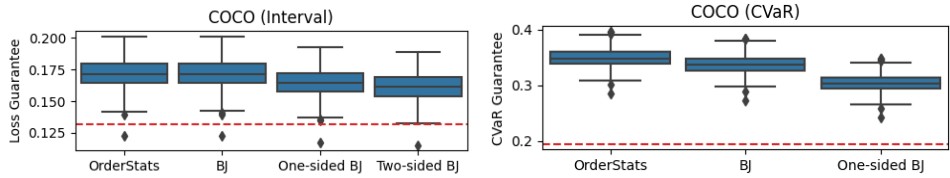

Figure 5: Results for bounding the quantities of the VaR for various methods. The average test loss over all trials is plotted in red.

where $\mu \in \mathbb{R}^5$ is a vector of class weights representing the importance of sensitivity for each test item's class label. This allows for user flexibility, where for instance a school could prefer to be more sensitive towards detecting good students. To demonstrate this, we perform our experiments with $\mu = (0.5, 0.6, 0.7, 0.8, 0.9)$.

We use 8,683 samples from the UCI Nursery train split, including examples from both groups, to train a logistic regression classifier to predict a prospective student's admission ranking. In our bounding experiments, 4,000 samples from the original test split are used for validation and testing, split evenly between the groups, with 500 examples from each group taken randomly for validation each trial. Point-wise methods are run with a beta grid size of 50. Since fairness is often concerned with those examples with the highest loss, we focus on bounding the CVaR in the range $[0.9, 1.0]$.

### E.6 ADDITIONAL RESULTS

#### E.6.1 VALUE OF OPTIMIZING FOR TARGET METRIC

Results for additional experiments highlighting the value of optimizing for a given target metric are shown in Table 5

| Dataset | Target | Mean | VaR | Interval | CVaR |
|---------|--------|------|-----|----------|------|
| CIFAR-100 | Mean | **0.156 ± 0.007** | 0.507 ± 0.009 | 0.507 ± 0.016 | 0.664 ± 0.009 |
| | VaR | 0.202 ± 0.024 | **0.411 ± 0.028** | 0.433 ± 0.020 | 0.651 ± 0.015 |
| | Interval | 0.200 ± 0.022 | 0.418 ± 0.031 | **0.429 ± 0.020** | 0.647 ± 0.017 |
| | CVaR | 0.353 ± 0.126 | 0.466 ± 0.038 | 0.468 ± 0.034 | **0.627 ± 0.007** |
| Go Emotions | Mean | **0.147 ± 0.007** | 0.503 ± 0.091 | 0.472 ± 0.057 | 0.700 ± 0.015 |
| | VaR | 0.178 ± 0.016 | **0.230 ± 0.023** | 0.312 ± 0.029 | 0.627 ± 0.030 |
| | Interval | 0.195 ± 0.021 | 0.257 ± 0.027 | **0.289 ± 0.026** | 0.589 ± 0.027 |
| | CVaR | 0.241 ± 0.044 | 0.314 ± 0.050 | 0.323 ± 0.045 | **0.570 ± 0.026** |

Table 5: CIFAR-100 and Go Emotions results illustrating trade-offs between targeting target different metrics in predictor selection and the guarantees that can be issued for each metric given by the selected bound. We use the Berk-Jones bound, and bold results are best for a given metric.

#### E.6.2 TRUNCATED BERK-JONES VS. EXISTING METHODS

Results for bounding a single predictor on MS-COCO are shown in Figure 5.

#### E.6.3 MS-COCO EXPERIMENTS ON ALL RISK MEASURES

All results for MS-COCO experiments comparing the performance of various bounding methods across the full canonical set of metrics (Mean, VaR, Interval, and CVaR) are reported in Table 6.

#### E.6.4 VIOLATIONS OF LOSS GUARANTEES AND LOWER CONFIDENCE BOUNDS

Tables 7 and 8 show violations of both the loss guarantee and the CDF lower confidence bound returned by each method for a subset of the MS-COCO experiments in Section 5.3 and the CIFAR-

|  | Method | Guarantee | Actual |  | Method | Guarantee | Actual |
|---|---|---|---|---|---|---|---|
| Mean | LttHB | 0.097 ± 0.004 | 0.048 ± 0.001 | Interval | DKW | 0.744 ± 0.005 | 0.146 ± 0.012 |
|  | RcpsWSR | **0.055 ± 0.003** | 0.048 ± 0.001 |  | OrderStats | 0.189 ± 0.011 | 0.138 ± 0.007 |
|  | KS | 0.142 ± 0.003 | 0.048 ± 0.001 |  | KS | 0.581 ± 0.008 | 0.144 ± 0.013 |
|  | BJ | 0.088 ± 0.003 | 0.048 ± 0.001 |  | BJ | 0.189 ± 0.011 | 0.138 ± 0.007 |
|  |  |  |  |  | One-sided BJ | 0.183 ± 0.010 | **0.137 ± 0.006** |
|  |  |  |  |  | Two-sided BJ | **0.182 ± 0.010** | **0.137 ± 0.006** |
| VaR | LttHB | 0.968 ± 0.036 | 0.154 ± 0.01 | CVaR | DKW | 1.0 ± 0.0 | 0.500 ± 0.000 |
|  | DKW | 0.287 ± 0.023 | 0.150 ± 0.020 |  | OrderStats | 0.454 ± 0.015 | 0.208 ± 0.011 |
|  | OrderStats | **0.166 ± 0.010** | **0.132 ± 0.004** |  | KS | 0.971 ± 0.002 | 0.228 ± 0.022 |
|  | KS | 0.287 ± 0.023 | 0.150 ± 0.020 |  | BJ | 0.453 ± 0.015 | 0.208 ± 0.011 |
|  | BJ | 0.179 ± 0.010 | 0.134 ± 0.006 |  | One-sided BJ | **0.419 ± 0.015** | **0.207 ± 0.010** |

Table 6: Results for experiments with MS-COCO Object Detection and all canonical metrics. Best results in bold, except for actual mean where all methods produced the same loss.

100 and Go Emotions experiments in Section 5.2. The experiment in Table 7 involves bounding a space of predictors, while that in Table 8 uses a single fixed predictor.

We can make a few observations from these results. First, specific loss measures are violated less frequently than CDF lower confidence bounds. A CDF violation occurs whenever the true CDF lies below the CDF-LCB at any point. But many risk measures average over a range of quantiles, or only focus on certain quantiles. In that case, it is possible for the CDF bound to be violated but not the risk measure guarantee. Next, specific loss distributions may be better behaved in practice than the theory accounts for. We deal with distribution-free methods, meaning that they hold for any possible loss distribution with only minimal assumptions (e.g. the loss is bounded). If certain properties of the loss distribution were known (for example, it belongs to a Beta distribution with unknown parameters), then we could in theory take this into account to produce tighter bounds. Since ours are distribution-free, there should be some distribution for which our bounds are tight, but it may not be the particular loss distribution we encounter. A similar phenomenon occurs with e.g. Hoeffding's inequality, where the bound is only tight when the probability mass is concentrated on the endpoints. Finally, bounding multiple predictors simultaneously affects the bounds, as shown by the difference in violation levels between Tables 7 and 8. Applying the union bound to multiple predictors assumes that they are independent. But in practice the predictors may be highly correlated, meaning that the ordinary Bonferroni correction is overly conservative.

| Metric | Method | Loss Viol. | LCB Viol. |
|---|---|---|---|
| VaR Interval | DKW | 0.000 | 0.000 |
|  | OrderStats | 0.001 | 0.004 |
|  | KS | 0.000 | 0.000 |
|  | BJ | 0.001 | 0.002 |
|  | One-sided-BJ | 0.001 | 0.007 |
|  | Two-sided-BJ | 0.001 | 0.006 |
| CVaR | DKW | 0.000 | 0.000 |
|  | OrderStats | 0.000 | 0.002 |
|  | KS | 0.000 | 0.000 |
|  | BJ | 0.000 | 0.003 |
|  | One-sided-BJ | 0.000 | 0.005 |

Table 7: Violations of loss guarantees and lower confidence bounds for MS-COCO dataset when using bounding methods to select a predictor with $\delta = 0.05$.

### E.7 PREDICTOR SPACE

For each experiment trial and random data split, we test a finite sample of 500 linearly spaced thresholds (or predictors) on the validation set. For computational efficiency, upper and lower bounds are

| Dataset | Method | Loss Viol. | LCB Viol. |
|---|---|---|---|
| CIFAR-100 | DKW | 0.000 | 0.000 |
| | OrderStats | 0.000 | 0.004 |
| | KS | 0.000 | 0.031 |
| | BJ | 0.000 | 0.020 |
| | One-sided-BJ | 0.000 | 0.026 |
| | Two-sided-BJ | 0.002 | 0.041 |
| Go Emotions | DKW | 0.000 | 0.001 |
| | OrderStats | 0.000 | 0.007 |
| | KS | 0.000 | 0.028 |
| | BJ | 0.000 | 0.021 |
| | One-sided-BJ | 0.003 | 0.041 |
| | Two-sided-BJ | 0.005 | 0.043 |

Table 8: Violations of loss guarantees and lower confidence bounds for CIFAR-100 and Go Emotions datasets when using a fixed predictor with $\delta = 0.05$.

set as the maximum and minimum values of the scores over the entire dataset, although this could easily be replaced by the maximum and minimum value of the scores on the validation set alone. Results are measured on the test data using the predictor with the best guarantee for a particular target metric.

