# OpenReview forum: "Quantile Risk Control: A Flexible Framework for Bounding the Probability of High-Loss Predictions"
_ICLR.cc/2023/Conference — ICLR 2023 poster_

### Official Review · Reviewer_Zm2X · 2022-10-25

**Confidence:** 3
**Correctness:** 4
**Technical Novelty And Significance:** 3
**Empirical Novelty And Significance:** 3
**Recommendation:** 6

**Clarity, Quality, Novelty And Reproducibility:**

No concerns. The paper is relatively easy to follow and seems to make novel contributions to an area of study that receives relatively little attention from the ML community today (or at least the part of it where I work), but arguably deserves more. Perhaps it is better-known in the finance and medical fields, where worst-case outcomes are relatively more important than average-case.

**Strength And Weaknesses:**

Strength:
* The technique is widely applicable for any quantile-based risk function and any set of black-box ML models that produce prediction functions.

Weaknesses:
* The 95% confidence bound on the CDF displayed in Figure 2 looks very conservative, given 1000 samples. I wonder whether there are alternate methods to produce tighter bounds on the CDF.
* Some of the bounds in the empirical section appear to be quite loose, with a 1.5 to 2x gap between the bound and the actual metric (e.g., Mean and CVaR in Table 2, and both results in Table 3). Whether this has practical implications would depend on whether the technique is being used to produce tight bounds on the metrics, or only to select the best model among several candidates.

**Summary Of The Paper:**

The paper introduces a framework for measuring and optimizing for quantile-based risk functions, which go beyond the standard practice of measuring and optimizing for the average (expected) loss function. The framework allows one to optimize for any quantity expressed as an integral over the quantile function of the loss; common examples include Value-at-Risk (VaR), Conditional Value-at-Risk (CVaR), and VaR-Interval. These quantile-based risk functions tend to be used to measure something closer to the worst-case loss, rather than the average loss. This is useful when planning to guard against tail risk in a setting where an ML model is deployed.

The paper introduces a novel way to define a lower confidence bound (LCB) on the loss function's cumulative distribution function (CDF), which equivalently produces an upper confidence bound on the loss function's quantile function (the quantile function is the inverse CDF), which produces an upper confidence bound on any quantile-based risk function.

**Summary Of The Review:**

Overall, I like this area of study, and I find the proposed technique to be interesting and potentially useful. I like that the technique can be used on any set of pre-trained, black-box models, and that it can be used to select the model that optimizes a given quantile-based functions of a loss. However, that the bounds that are produced seem quite loose in some cases, which makes me question whether practitioners would actually trust that the method is giving useful (tight) bounds.

---

> ### Author Response · Authors · 2022-11-18
> **Response to Zm2X**
>
> Thank you for your thoughtful and overall positive feedback. Here, we provide answers to your concerns point by point.
>
> **Q1:** The $95\\%$ confidence bound on the CDF displayed in...
>
> **A1:** Please note that the bound in Figure 2 was formed with $n=10$ samples.  We have updated the figure and caption, and hope it is clear now.
>
> **Q2:** Some of the bounds in the empirical section appear to be quite loose...
>
> **A2:** With regards to the tightness of the bounds, we have updated the experiments section to highlight the tightness of the truncated Berk-Jones bounds and address the reviewer’s concerns.  The boxplots in Section 5.2 show that the truncated Berk-Jones bounds can be close to the empirical CDF, and are significantly tighter than those given by other methods.  We also show in Section 5.4/Appendix that the CDF violations for our method are tightest to the target level $\delta$, and do approach $\delta$ in the case where other statistical corrections are not being applied.

---

> ### Author Response · Authors · 2022-11-23
> **A gentle ping**
>
> Hi - Now that the discussion period has ended we thought we would ping you just to check to see if we addressed your comments and questions to your satisfaction in our response and revised manuscript. We know it is a very busy time of year, but please let us know if you
> have any other questions or comments.

---

### Official Review · Reviewer_yK7n · 2022-10-26

**Confidence:** 3
**Correctness:** 4
**Technical Novelty And Significance:** 3
**Empirical Novelty And Significance:** 2
**Recommendation:** 5

**Clarity, Quality, Novelty And Reproducibility:**

Clarity
The paper is well written overall.

Quality + Novelty
The framework proposed is novel, but the accompanying theoretical results lack novelty.

Reproducibility
The authors mention that they will release code on github.

**Strength And Weaknesses:**

Strength
The framework produced by the paper generalizes exisitng known frameworks for quantifying uncertainty in classifier predictions. The paper is quite well written and easy to follow, and the arguments are supported by both theoretical and empirical results.

Weakness
While the paper doe sa good job of laying the foundations of the framework, it does not stand out on from either the theoretical or the empirical perspective.
- From a theoretical standpoint, most of the theoreems (1, 2, 3) are simple extensions of known results on CDFs and concentration inequalities. There is no discussion on what the challenges were in proving these results, and while they are essential to establish the framework, they do not provide new novel insights.
- From the empirical side of things, this framework can be quite impactful. Currently, the paper focusses on results for COCO object detection and UCI datasets. There is very little difference in the actual performance when using any of the methods in the literature (Table 2, 3) - why is this the case?
- Quantile losses are typically used when data is very noisy (say finance time series) or when there are robustness issues to be dealt with. It looks like some of the methods might be more useful in those scenarios and trying out the proposed methods for such applications would strengthen the paper.


**Summary Of The Paper:**

The paper proposes a new framework for obtaining bounds on quantiles of the loss distribution of a given predictor. The method uses a lower bound on the CDF to obtain high-confidence bounds on the quantile loss profiles of the predictor.

**Summary Of The Review:**

The proposed framework for quantile loss uncertainty estimation is quite nice, but the paper needs to be updated to showcase the empirical utility of this framework.

---

> ### Author Response · Authors · 2022-11-18
> **Response to Reviewer yK7n**
>
> Thanks for your thoughtful feedback. We will address your comments point by point below, and we hope that our answers can ease your concern.
>
> **Q1:** From a theoretical standpoint, most of the theorems (1, 2, 3) are simple extensions of known results on CDFs and concentration inequalities.
>
> **A1:** We respond to these points here. We also added a paragraph highlighting our innovation in the revised PDF (see paragraph ``Novelty of QRC").
>
> (i). Novelty of the framework: the overall framework that uses nonparametric CDF lower confidence bounds as an underlying representation to simultaneously bound any quantile based risk minimization (QBRM). Previous works in distribution-free uncertainty quantification have targeted the mean and VaR, but to our knowledge none have focused on either the CVaR or VaR-interval.
>
> (ii). Finite sample bounds: as pointed out by the reviewer, our theorems are based on extensions of existing results, but those results are mainly asymptotic (not for finite sample size), and bridging the gap from asymptotic results to non-asymptotic ones has great significance in distribution-free uncertainty quantification. Specifically, even though some literature talks about bounding QBRMs, for instance, in [1], they mainly show how to bound QBRMs given a CDF estimate, they do not provide the tools to bound the CDF from a finite data sample. Our framework build non-asymptotic results by leveraging modern goodness-of-fit test statistics to produce high-quality nonparametric CDF lower bounds from finite data, and applying them to bounding QBRMs for multiple predictors simultaneously.
>
> (iii). New techniques: We highlight that we propose truncated Berk-Jones statistics, which refocus the statistical power of standard Berk-Jones to target either the CVaR (one-sided) or VaR-interval (two-sided).
>
> **Q2:** From the empirical side of things, this framework can be quite impactful. Currently, the paper focusses on results for COCO object detection and UCI datasets. There is very little difference in the actual performance when using any of the methods in the literature (Table 2, 3) - why is this the case?
>
> **A2:** We have significantly updated the experimental sections of the paper to highlight the empirical usefulness of the QRC framework and our novel bounds on intervals of the VaR and CVaR.  As mentioned in our General Responses above, we have reorganized the experimental sections (please see Section 5 in the revised PDF). In summary, first we show in 5.1 that in order to issue a strong guarantee for a given target risk measure, it is key to optimize for that metric, and thus it is important to have strong bounds for quantities like intervals on the VaR and CVaR.  Next, in the updated Section 5.2 we compare our novel truncated Berk-Jones bounds to other baseline methods for bounding these quantities using a finite data sample, and show that the truncated Berk-Jones bounds clearly give the best guarantees and are closest to the empirical loss.  With regards to the empirical loss induced by the predictor chosen using each bound, in general, the best guarantees have best actual loss.  The difference can be small (or none) though, as all methods may choose the same predictor or similar predictors around that which minimizes the target risk measure on the validation set.
>
> **Q3:** Quantile losses are typically used when data is very noisy ... It looks like some of the methods might be more useful in those scenarios and trying out the proposed methods...
>
> **A3:** To address the reviewer’s concerns regarding the performance on noisy data, we have our updated experiments section to highlight the strength of the novel truncated Berk-Jones bound on tail quantities like the CVaR.
>
>
> [1] Kevin Dowd. Using Order Statistics to Estimate Confidence Intervals for Quantile-Based Risk Measures.

---

> > ### Author Response · Authors · 2022-11-22
> > **A Gentle Ping**
> >
> > Hi - Now that the discussion period has ended we thought we would ping you just to check to see if we addressed your comments and questions to your satisfaction in our response and revised manuscript. We know it is a very busy time of year, but please let us know if you
> > have any other questions or comments. Thanks in advance!

---

### Official Review · Reviewer_xSdZ · 2022-10-29

**Confidence:** 3
**Correctness:** 4
**Technical Novelty And Significance:** 3
**Empirical Novelty And Significance:** 3
**Recommendation:** 6

**Clarity, Quality, Novelty And Reproducibility:**

The overall clarity, and quality is good. However, some figures are hard to read: can you make the axes and legend bigger in Figure 2? And maybe different contrasting colors could be used for Figure 3 left, since green and blue both appear muddled with short boxes.

**Strength And Weaknesses:**

The paper is coherent, well-written and addresses an interesting and seemingly novel problem of bounding quantile-based measures of risk (as opposed to the usual definition of risk, which takes the expectation). The arguments made in developing the proposed method seems sound.

Regarding the empirical results, however, I'm not sure if it tells a convincing case for the usefulness of the bounds.
First off, there's not much of a comparison to be done here, since the proposed method of bounding QBRM's is applied with all methods (i.e. test statistics), and they all seem to have close to 0 violation of their guarantees. The newly proposed Truncated-BJ statistic does tend to produce tighter bounds. What was the alpha value used? Is 0% violation actually expected, or are all of the bounds too loose? Also, it's also difficult to tell how significant the difference is between the upper bound and the actual values for each metric. Can you scan different values of alpha? Lastly, if using a specific test statistic and applying the proposed QBRM bound leads to a bound that is way too loose, would it be considered a fault of the test statistic or the bounding method?


**Summary Of The Paper:**

This work proposes a method to bound a notion of risk that is measured in terms of quantiles, called "quantile-based risk measures (QBRM)". For a given predictor, a distribution over loss values is induced, and this paper shows that a lower bound of the CDF of this distribution serves as an upper bound of any QBRM of this predictor. They further show that it's possible to uniformly lower bound the CDF of the loss distribution for a set of predictors, which further serves as an upper bound to any QBRM for all predictors, especially the optimized predictor. The CDF bound can be derived via various minimum-type test statistics. Empirical results compare the tightness of the bounds produced by a suite of test statistics, various QBRM's, and demonstrate the guarantees are generally met.

**Summary Of The Review:**

While I think this paper addresses an interesting and relevant problem and proposes a sound method, it's also a little difficult to tell the significance of the empirical results. I am looking forward to the author's response.

---

> ### Author Response · Authors · 2022-11-18
> **Response to Reviewer xSdz**
>
> Thank you for your constructive suggestions and overall positive feedback. Here, we provide answers to your concerns point by point.
>
> **Q1:** Regarding the empirical results, however, I'm not sure if it tells a convincing case for the usefulness of the bounds...
>
> **A1:** To address your points, we have reorganized the experiments section (please also see General Responses above).  We include an extensive discussion and further results for the violations of the loss guarantees and CDF bounds.  We show that the CDF violations for our method are close to the target level $\delta$ when the bound is not subject to other statistical corrections, and make a few key observations.  First, CDF lower confidence bounds are violated more frequently than specific loss measures, which is what we reported in the original tables.  Next, specific loss distributions may be better behaved in practice than the theory accounts for.  Since our methods are distribution-free, there should be some distribution for which our bounds are tight, but it may not be the particular loss distribution we encounter. Finally, bounding multiple predictors simultaneously affects the bounds.  Applying the union bound to multiple predictors assumes that they are independent, but in practice the predictors may be highly correlated, meaning that the ordinary Bonferroni correction is overly conservative.
>
> We believe that our new experiments section further highlights the strength of the truncated BJ bounds, especially in tail quantities like the CVaR.  We have also produced a new Figure 2 and re-produced the box plots to only include the key quantity, which is the guarantee, which we hope makes them easier to understand.

---

> ### Author Response · Authors · 2022-11-23
> **A gentle ping**
>
> Hi - Now that the discussion period has ended we thought we would ping you just to check to see if we addressed your comments and questions to your satisfaction in our response and revised manuscript. We know it is a very busy time of year, but please let us know if you
> have any other questions or comments.

---

### Official Review · Reviewer_Vv5L · 2022-11-04

**Confidence:** 5
**Correctness:** 4
**Technical Novelty And Significance:** 3
**Empirical Novelty And Significance:** 2
**Recommendation:** 6

**Clarity, Quality, Novelty And Reproducibility:**

The paper is clearly written. The significance could be emphasized further, though I believe it is there (see weaknesses for details). The results appear to be easily reproducible.

**Strength And Weaknesses:**

### Strengths

* Characterizing the performance of a predictor in the extremes, in addition to in average, can help mitigate risks. This is particularly important for life-threatening or society-disturbing applications of machine learning. This paper addresses this directly by formulating it as a characterization and control of the quantiles of the risk.

* The general methodology provided to derive CDF bounds based on goodness-of-fit statistics provides many useful insight. In particular, the benefits of finer bounds deeper in the tail are evident, especially for integral-type quantile metrics where inaccuracies accumulate. In general, the paper opens up opportunities to bring more ideas from statistics into machine learning.

 * The paper is organized and written very well.

### Weaknesses

* The use of quantiles to control the risk of predictors is not in itself new, as it is shared by many uncertainty quantification techniques. Order statistics always naturally show up in this context. The technical contributions are also quite straightforward, since connections between CDFs and quantiles through inversion is standard. However, this is counterbalanced by the clarity of combining the many ideas, and the emphasis on some of the new insights, like the use of the truncated Berk-Jones statistics to get finer bounds in the tail.

* The significance of the contributions is a bit debatable. It is not clear how much better the novel aspects of this methodology help. For example, my first insight when I started reading the paper was to just use the empirical process lower bounds, i.e., the Dvortesky-Kiefer-Wolfowitz-Massart inequality, by capping it left and right. It tourns out the paper considers this as one of the baselines, and it does perform mostly equivalently to the best proposed approach. I think examples where the new contributions show more striking gains (which I believe exist, since these should do much better in the tail) can help bring out more of the significance of this work. For instance, it is worth elaborating more on when integral metrics like CVaR are most appropriate, since this is where the new stuff shines.

* Generally, it feels that too much of the core contributions currently resides in the appendix. This may be okay for proofs, but when it comes to experimental results, they should be in the main text (namely I’m talking about the fairness experiments results.)


**Summary Of The Paper:**

The paper proposes a method to characterize, at validation time, the extreme deviation of a predictor and possibility using this characterization to select among a subset of these predictors, e.g., by adjusting a threshold. The deviation is measured through notions such as value-at-risk, i.e., the value of a given high quantile. The method consists of lower bounding the CDF of the risk with high probability. Experiments are given to illustrate the use of the technique.

**Summary Of The Review:**

The paper bridges many interesting ideas, even if not majorly novel. The significance of the results is a bit weak, as some straightforward approaches can compete with it. That said, the paper is an insightful contribution to the efforts of characterizing and mitigating risks of predictors in machine learning, and may be worthwhile to share with the community.

---

> ### Author Response · Authors · 2022-11-18
> **Response to Reviewer Vv5L**
>
> Thank you for your thoughtful and overall positive feedback. Here, we provide answers to your concerns point by point.
>
> **Q1:** The use of quantiles to control the risk of predictors is not in itself new... However, this is counterbalanced by the clarity of combining the many ideas, and the emphasis on some of the new insights...
>
> **A1:** We'd like to point out the two primary novel aspects of our framework.
>
> The first is the overall framework, which uses nonparametric CDF lower confidence bounds as an underlying representation to simultaneously bound any quantile based risk minimization (QBRM). Previous works in distribution-free uncertainty quantification have targeted the mean and VaR, but to our knowledge none have focused on either the CVaR or VaR-interval. In addition, even though order statistics are frequently used in contexts involving CDFs, they are primarily used in hypothesis testing, where most results are asymptotic (not for finite sample size). Also, while others have shown how to bound QBRMs given a CDF estimate [1], they do not provide the tools to bound the CDF from a finite data sample. Our framework brings these two ideas together by leveraging modern goodness-of-fit test statistics to produce high-quality nonparametric CDF lower bounds from finite data, and applying them to bounding QBRMs for multiple predictors simultaneously.
>
> The second novel aspect of QRC is our proposed truncated Berk-Jones statistics, which refocus the statistical power of standard Berk-Jones to target either the CVaR (one-sided) or VaR-interval (two-sided). We add a paragraph discussing our innovation in the revised PDF (see paragraph ``Novelty of QRC").
>
> **Q2:** The significance of the contributions is a bit debatable...I think examples where the new contributions show more striking gain...
>
> **A2:** Please refer to A1 above for details about the significance of the contributions of our framework. In addition, for the expected gain of our proposed-best-approach, please refer to the General Responses discussion of the new and improved Experiments section. As the reviewer expected should be the case, the truncated Berk-Jones bound gives the biggest improvement for bounding CVaR and other tail quantities, and we believe our current set of experiments better highlight this.
>
> **Q3:** ...Generally, it feels that too much of the core contributions currently resides in the appendix...
>
> **A3:** To address your concern, we have significantly reorganized our experimental results and added new sets of results. All of our main experimental results are now included in the main paper.
>
>
> [1] Kevin Dowd. Using Order Statistics to Estimate Confidence Intervals for Quantile-Based Risk Measures.

---

> ### Author Response · Authors · 2022-11-23
> **A gentle ping**
>
> Hi - Now that the discussion period has ended we thought we would ping you just to check to see if we addressed your comments and questions to your satisfaction in our response and revised manuscript. We know it is a very busy time of year, but please let us know if you
> have any other questions or comments.

---

### Author Response · Authors · 2022-11-18
**General Responses to all Reviewers**

We thank all four reviewers for their constructive and positive feedback, and for thinking our paper provides new insight, addresses important applications, is organized and well-written.

We have revised the draft according to the feedback, with major changes highlighted below:

**Revised method sections.** We significantly revised the presentation of the method to make it clearer and easier to read (see Sections 4 and 5 in the new PDF). We also highlight our main contributions. For example, in Section 4 of the revised PDF --- Quantile Risk Control (QRC), we present the outline of our method and list our innovations.

**Revamped Experiments section.** We have reorganized the Experiments section in order to highlight the significance of our method.  We do this in two steps.  First, we show that in order to issue a strong guarantee for a certain metric, it is key to optimize for that metric.  Thus, while there are strong existing methods for bounding quantities like the mean and VaR while optimizing over a predictor space, it is important to have strong bounds for quantities like intervals on the VaR and CVaR.  Next, we compare our novel truncated Berk-Jones bounds to other baseline methods for bounding these quantities using a finite data sample.  In this case, the truncated Berk-Jones bounds provide the best guarantees and are closest to the empirical loss.

---

> ### Author Response · Authors · 2022-11-18
> **Code Submission**
>
> We have uploaded our code under supplementary materials.  Thank you again for your time and feedback.

---

### Decision · Program_Chairs · 2023-01-20

**Decision:**

Accept: poster

**Justification For Why Not Higher Score:**

Reviewers have not given the submission very high scores.

**Justification For Why Not Lower Score:**

Very solid research with clear potential benefit for follow-on research and ultimately applications.

**Metareview: Summary, Strengths And Weaknesses:**

(a) The paper shows how to provide mathematical guarantees about the tail of the distribution of losses incurred when making predictions, i.e., just the worst predictions.

(b) High quality mathematical work, clear relevance to fairness in ML applications.

(c) Reviewers who are experts are lukewarm about the mathematical novelty. The actual computational benefit is small (Table 2).

An ICML 2022 workshop contained the paper "VaR-Control: Bounding the Probability of High-Loss Predictions" by Jake Snell, Thomas Zollo and Richard Zemel.

Comment from the AC: The application to fairness needs more emphasis. The nursery school dataset is from the 1980s and the meaning of "convenient" for financial status is unclear. Moreover, it is unclear what the fairness goal is here; maybe poorer families should be prioritized? Or maybe richer families must be prioritized in order for the nursery school to be viable financially? (The authors' university has the luxury of doing need-blind admissions, but other colleges are less privileged.)

**Note From Pc:**

if the above contains the word "oral" or "spotlight" please see: "oral" presentation means -> notable-top-5% and "spotlight" means -> notable-top-25%. As stated in our emails, we are disassociating presentation type from AC recommendations

**Summary Of Ac-Reviewer Meeting:**

No meeting.